# Histologic Evidence of Oral and Periodontal Regeneration Using Recombinant Human Platelet-Derived Growth Factor

**DOI:** 10.3390/medicina59040676

**Published:** 2023-03-29

**Authors:** Mohamed M. Meghil, Obada Mandil, Myron Nevins, Muhammad H. A. Saleh, Hom-Lay Wang

**Affiliations:** 1Department of Periodontics, Dental College of Georgia, Augusta University, Augusta, GA 30912, USA; 2Department of Periodontics and Oral Medicine, School of Dentistry, University of Michigan, Ann Arbor, MI 48109, USA; 3Division of Periodontology, Department of Oral Medicine, Infection and Immunity, Harvard School of Dental Medicine, Boston, MA 02115, USA

**Keywords:** periodontal regeneration, bone regeneration, gingival recession, regenerative medicine, platelet-derived growth factor (PDGF)

## Abstract

Human histology provides critical information on the biological potential of various regenerative protocols and biomaterials, which is vital to advancing the field of periodontal regeneration, both in research and clinical practice. Outcomes of histologic studies are particularly valuable when interpreted considering additional evidence available from pre-clinical and clinical studies. One of the best-documented growth factors areproven to have positive effects on a myriad of oral regenerative procedures is recombinant human platelet-derived growth factor—BB (rhPDGF-BB). While a systematic review of clinical studies evaluating rhPDGF in oral regenerative procedures has been recently completed, a review article that focuses on the histologic outcomes is needed. Hence, this communication discusses the histologic effects of rhPDGF-BB on oral and periodontal regenerative procedures, including root coverage and soft tissue augmentation, intrabony defects, furcation defects, peri-implant bone augmentation, and guided bone regeneration. Studies from 1989 to 2022 have been included in this review.

## 1. Introduction

Periodontal tissue engineering requires three basic elements: cells, scaffolds/matrices, and signaling molecules (growth factors) [1]. Cells such as mesenchymal stem cells, osteoprogenitor cells, and periodontal ligament fibroblasts can migrate from the surrounding host tissues into the scaffold under the influence of chemoattractants and mediators such as growth factors [2]. Scaffolds are the foundation that prevents tissue collapse during healing, stabilize blood clotting, and provide essential support for cell attachment and proliferation. Growth factors, through their chemotactic and mitogenic signals, induce cell migration and proliferation, respectively, in the defect site. Growth factors have been the focus of extensive dental research due to their robust functions, their importance in tissue regeneration, and the possibility of using them to improve grafts that act primarily as scaffolds. Based on all these reasons, growth factors have been extensively used in the field of periodontal regeneration and healing.

Growth factors are molecules secreted by cells and are capable of stimulating a variety of cellular activities, including cell proliferation, migration, differentiation, and multicellular morphogenesis during both embryonic development and tissue healing and repair. Identified in the 1970s as a serum growth factor for multiple types of cells [3,4,5], platelet-derived growth factor (PDGF) and its receptors (PDGFRs) have served as prototypes for growth factor function and research for more than 40 years. PDGF has four isoform homodimers, AA, BB, CC, and DD, in addition to a heterodimer, PDGF-AB. The receptors for PDGF dimers on cell surfaces have tyrosine kinase cytoplasmic domains and are known as PDGFRα and/or PDGFRβ. The receptors dimerize before binding to different isoforms of PDGF. While PDGF-AA, -BB, and -CC bind to PDGFR α/α, PDGF-AB, -BB, -CC, and -DD bind to PDGFR α/β [6]. In addition, PDGF-BB and DD bind to PDGFR β/β (Figure 1A) [6]. Because of these variations in the binding capabilities of PDGF homodimers and heterodimers to their dimerized receptors, the biological effect of PDGF depends on the expression level of PDGFR dimer on target cells. The most biologically active isoform of PDGF on periodontal and bone cells in the oral cavity is PDGF-BB.

At injury sites, platelets in the blood clot release α-granules containing PDGFs, which in turn attract and induce chemotaxis and the proliferation of a multitude of cells important in healing and bone regeneration, including periodontal ligament cells, mesenchymal stem cells (MSCs), and osteoprogenitor cells, as well as neutrophils and macrophages [7]. The latter play a crucial role in the initial steps of wound healing not only through their phagocytic activity but also as a continuous source of PDGFs and other growth factors [8]. The MSCs subsequently differentiate into osteoblasts [9,10]. In addition to its direct effects, PDGF-BB indirectly contributes to improved healing and bone regeneration by increasing the expression of angiogenic molecules essential for healing and bone regeneration, such as vascular endothelial growth factor (VEGF) (Figure 1B) [11,12].

Within the context of this manuscript, the effects of PDGF-BB on periodontal regenerative procedures, namely root coverage and soft tissue augmentation, intrabony defects, furcation defects, peri-implant bone augmentation, and guided bone regeneration, are summarized from studies that evaluated the histologic evidence in humans and animal models (Table 1). Studies from 1989 to 2022 have been included in this review. To our knowledge, this is the first review article that summarizes all the studies that have reported on the histologic evidence of the regenerative potential of PDGF.

## 2. The Effects of PDGF-BB on Periodontal Regenerative Procedures

### 2.1. Histological Outcomes in Intrabony Defects

Lynch et al. were the first to evaluate the effects of PDGF on periodontal and peri-implant regeneration. In a series of studies, they compared the histologic outcomes of open flap debridement to PDGF alone or in combination with insulin-like growth factors in terms of the regeneration of the soft and hard tissues of the periodontium in beagle dogs with naturally occurring periodontal disease [13,34]. Their studies found that periodontal defects treated with open flap debridement healed by the long junctional epithelium as expected. They observed that the LJE formed within two weeks of surgery in this model, whereas PDGF treated sites exhibited robust cementogenic and osteogenic activity within the same two-week period. They further reported the presence of an osteogenic front characterized by a dense population of osteoblastic cells within two weeks of treatment with PDGF. In another study, they found that while the OFD sites again showed epithelium migration apically to the base of the defect and an absence of new cementum formation on the adjacent root surface at five weeks following surgery, the sites treated with recombinant human PDGF-BB (rhPDGF-BB) again contained robust new bone formation, with osteocytes already incorporated into the newly formed osteoid by five weeks after treatment. Furthermore, the new bone was lined with osteoblast-like cells. Following five weeks of healing, the histometric analyses demonstrated that a combination of rhPDGF-BB and IGF-I significantly increased the bone height and density and length of new cementum compared to OFD, in some cases by as much as 5 mm supra-crestally. In addition, the newly formed bone contained numerous osteocytes and was lined by a continuous layer of osteoblasts. Of note, the PDGF treatment resulted in the physiologic regeneration of the normal architecture of the periodontium with new cementum, new bone, and a highly organized periodontal ligament running perpendicularly between the new bone and cementum.

Furthermore, the influence of PDGF-BB on the initial periodontal cellular response was investigated by Wang et al. on a mongrel dog model over 1, 3, and 7 days [14]. A closed-wound surgical model was made by creating 4 × 4 mm surgical fenestration defects in the dentin on the facial surface of the posterior teeth in each quadrant. The study compared the histological outcomes of filing the defect with either saline, expanded polytetrafluoroethylene (ePTFE) membrane, PDGF, or ePTFE + PDGF. The results demonstrated that PDGF significantly increased fibroblasts, cemetoblasts, osteoblasts, perivascular cells, and endothelial cells proliferation from day 1 to day 7. Interestingly, the study failed to achieve statistically significant differences between PDGF and ePTFE + PDGF, suggesting that the ePTFE membrane did not enhance the results of PDGF.

Sculean and coworkers studied the effect of PDGF on the regenerative potential of the connective tissue fibers of the periodontium [15]. The study included creating intrabony defects on either the mesial or distal aspects of the teeth of one monkey that was allowed to heal for 3 months. After that, sites were re-entered by raising full-thickness mucoperiosteal flaps, and notches were made on the root surface to mark the base of the bony defect. All the defects were then filled up with a gel containing PDGF, and only half of them were covered with bioabsorbable membranes. After a 5-month healing period, the histological analysis indicated the regeneration of the periodontal ligament fibers coronal to the notch that was created on the root surface. In addition, those fibers were inserted into a newly formed cementum, and newly regenerated alveolar bone was noticed. In addition, the regenerated oxytalan fibers were found to morphologically resemble those of the normal periodontal ligament.

Giannobile [35] and co-workers further characterized the biological effects of rhPDGF-BB or IGF-I individually and in combination in non-human primates and compared the effects to those of open flap debridement. At both 4 and 12 weeks, OFD-treated lesions generally revealed minimal osseous defect fill (ODF) (8.5 ± 2.1% and 14.5 +/− 5.7%, respectively) and minimal new attachment (NA) (34.1 +/− 5.2% and 26.6 +/− 10.5%, respectively). IGF-I treatment alone also did not significantly alter healing compared to a vehicle in any parameter at both 4 and 12 weeks. PDGF-BB-treated sites exhibited significant (*p* < 0.05) regeneration of NA (69.6 + 12.0%) at 12 weeks. Treatment with PDGF-BB/IGF-I resulted in 21.6 +/− 5.1% and 42.5 +/− 8.3% ODF at 4 and 12 weeks, respectively, and 64.1 +/− 7.7% and 74.6 +/− 7.4% NA at 4 and 12 weeks, respectively (all significantly greater than OFD, *p* < 0.05). The results from this study demonstrated that: (1) OFD does not result in new attachment or periodontal regeneration in non-human primates; (2) IGF-I alone also did not significantly alter periodontal wound healing; (3) PDGF-BB alone did significantly stimulate NA, with strong trends for promoting bone fill; and (4) the PDGF-BB/IGF-I combination resulted in significant increases in NA and bone defect fill above OFD at both 4 and 12 weeks. This study led to the focus of further research on the use of rhPDGF-BB (without IGF-I) with various bone graft materials.

Following those observations on animal models, Nevins and co-workers tested the effectiveness of three different concentrations of recombinant PDGF (rhPDGF-BB) (0.3 mg/mL, 1.0 mg/mL, and 5.0 mg/mL) on individuals with periodontal disease for the first time [16]. Teeth selected for this study were deemed hopeless or indicated for extraction due to other reasons to allow for histological analysis after extractions. Notches were created on the root surface of the involved teeth at the apical extent of the calculus. The osseous defects were treated with rhPDGF-BB combined with either demineralized freeze-dried bone allograft (DFDBA) or inorganic bovine bone in collagen (ABB-C), half of which was covered with a collagen membrane. After 9 months of healing, the histological outcomes demonstrated that rhPDGF-BB resulted in the regeneration of the periodontal attachment with the formation of new cementum, PDL, and bone. Strikingly, the apical extension of junctional epithelium (JE) stopped coronal to the newly formed bone even though no membrane was used.

Human histological studies were subsequently performed to provide further evidence for periodontal regeneration in intrabony defects by testing different concentrations of PDGF. A study evaluated the histological outcomes of the combination rhPDGF-BB + b-TCP in the treatment of intrabony defects in eight periodontally-diseased patients [17]. Two different concentrations of rhPDGF-BB were compared in this study to treat two teeth per patient, where one tooth was treated with 0.3 mg/mL of rhPDGF-BB + b-TCP, and the other tooth was treated with 1.0 mg/mL of rhPDGF-BB + b-TCP. A notch was created on each root surface, and the roots were treated with 50 mg/mL tetracycline. Histomorphometric analysis of biopsy sections after 6 months of healing showed that both concentrations of rhPDGF-BB resulted in the regeneration of new bone, new cementum, and a new periodontal ligament coronal to the notches. It is noteworthy that the study reported that newly formed cementum was observed on dentin and on old cementum. The presence of residual b-TCP particles was suggested to reduce the amount of new bone formation.

In 2010, Shirakata et al. evaluated basic fibroblast growth factor (bFGF), enamel matrix derivative (EMD), rhPDGF-BB + b-TCP, and control of open flap debridement (OFD) in the treatment of two-wall intrabony defects in beagle dogs, where each defect was treated with one of the treatments in each dog for 8 weeks [18]. The results showed that there were no significant differences in the histometric analysis between the bFGF and rhPDGF-BB + b-TCP groups, with both groups being superior to the EMD or OFD groups. In addition, rhPDGF-BB + b-TCP resulted in the formation of new bone along the root surface and new cellular cementum by inserting collagen fibers at the apical portion of the defects that changed to acellular extrinsic fiber cementum at the coronal portion. No new bone was observed on the small b-TCP remnants. In another study on beagle dogs, Zhang et al. tested the effectiveness of a growth factor local delivery system made of mesoporous bioglass (MBG)/silk fibrin scaffolds loaded with BMP7 and/or PDGF-B adenovirus [36]. Intrabony buccal dehiscent defects were created and filled with one of the following treatments: a control of no scaffold, MBG/silk scaffold alone, scaffold + adPDGF-B, scaffold + adBMP7, scaffold + adPDGF-B + adBMP7. After 8 weeks, the use of adPDGF-B resulted in significant regeneration of new cementum, bone, and periodontal ligament, by inserting new Sharpey’s fibers. Even though adBMP7 showed improved new bone formation, it did not result in significant new periodontal ligament formation.

### 2.2. Histological Outcomes in Furcation Defects

Histological evaluation of rhPDGF-BB in molar class II furcation defects was performed by Nevins and co-workers in the study that was discussed earlier in this review [16]. Histological evaluation showed regeneration of the periodontal attachment apparatus (new cementum, bone, and PDL) coronal to the reference notch. The new cementum was continuous on the root surface across the fornix. In addition, well-organized PDL with collagen fibers was inserted into the new cementum.

A clinical study on humans by Camelo et al. evaluated the histological outcomes of rhPDGF-BB combined with DFDBA for the treatment of class II furcation defects [19]. Two concentrations of rhPDGF-BB were tested in this study—0.5 mg/mL and 1.0 mg/mL. After 9 months, histologic evaluation of the En bloc biopsies revealed that all cases showed periodontal regeneration coronal to the reference notches that were made at the base of the calculus during the surgery. The newly formed bone was a mixture of lamellar and woven bone of the same density as the native bone. Abundant osteocytes embedded in the woven bone and normally-formed Haversian systems were also reported. The new cementum was continuous from the base of the notch to the fornix. The regenerated PDL was well-organized, with collagen fibers inserted from the bone into the newly formed cementum. Furthermore, no LJE was formed even though no membranes were used in the study. Strikingly, one of the cases showed a new cementum-like calcified matrix by inserting collagen fibers formed over enamel projections within the furcation. This is believed to be the first documented periodontal regeneration to form over an enamel projection in the fornix of a furcation.

Human histological studies were subsequently performed to provide further evidence for the periodontal regeneration of class III furcation defects using rhPDGF-BB/b-TCP [20]. Four patients with one through-and-through furcation defect in each patient were treated with rhPDGF-BB (0.3 mg/mL) + β-TCP, and the defects were covered with collagen membranes. After a 6-month healing period, the results showed that most sites had at least partial periodontal regeneration with new cementum, bone, and PDL. A lack of soft tissue attachment to the roof of the furcation in one site was also observed, and another site showed a surface healed with LJE. These sites were reported to have more extensive bacteria plaque accumulation and at least grade II mobility.

### 2.3. Histological Outcomes in Root Coverage Procedures and Soft Tissue Augmentation

A split-mouth, randomized clinical trial by McGuire et al. compared coronally advanced flap with rhPDGF-BB + β-TCP (-tricalcium phosphate) + bioabsorbable collagen wound dressing and coronally advanced flap with connective tissue graft [30]. They tested the regenerative potential of both groups using surgically created gingival recession defects performed on teeth requiring orthodontic extractions. The histological analysis of the En bloc sections after 9 months follow-up revealed that the CTG-treated sites’ healing occurred by the formation of a long junctional epithelium (LJE) with no evidence of true periodontal regeneration. Despite the formation of abundant connective tissue, the direction of the collagen fibers was parallel to the root surface rather than perpendicular and inserting into it, and therefore the presence of Sharpey’s fibers was missing. On the contrary, the rhPDGF-BB + β-TCP group showed signs of true periodontal regeneration evidenced by the formation of well-developed periodontal ligament (PDL), new cementum, and new bone. Furthermore, the study demonstrated that the connective tissue fibers of the newly regenerated PDL were oriented perpendicular to the new cementum and new bone, forming a regenerated PDL with new Sharpey’s fibers.

Simion et al. then tested the effect of using a rhPDGF + collagen matrix (CM) on soft tissue augmentation around dental implants in the esthetic region [31]. Following a 4-month healing period, the histological evaluation of the harvested soft tissue biopsy revealed not only an architecture of the regenerated soft tissue resembling that of the healthy gingival mucosa, but also complete resorption of the collagen matrix.

### 2.4. Histological Outcomes in Peri-Implant Regenerative Therapy

An evaluation of rhPDGF in ridge augmentation around dental implants was performed in a proof-of-concept study conducted by Simion et al. in a canine model [32]. In this study, surgically created vertical ridge defects were grafted with a deproteinized bovine bone block covered with a collagen membrane, a deproteinized bovine bone block infused with rhPDGF-BB, or a deproteinized bovine bone block infused with rhPDGF-BB, and covered with a collagen membrane (Figure 2A,B). The deproteinized bovine bone blocks were stabilized in the bony defects by means of two titanium implants. After 4 months, histologic examination showed that, while the deproteinized bovine bone block plus a collagen membrane resulted in little or no bone regeneration in this challenging model, the combination of rhPDGF-BB-infused deproteinized bovine bone and collagen membrane had significant amounts of bone regeneration. The best results, however, were found in the sites treated with rhPDGF-BB-infused deproteinized bovine bone without a collagen membrane. The latter sites contained a significant amount of new bone formation and bone-to-implant contact across the entire length of the implant. Later, another canine study by the same group tested the combination of rhPDGF-BB with hydroxyapatite and a collagen (eHAC) block [33]. This combination was compared to using the eHAC block alone, the eHAC block + collagen membrane, and the eHAC block + rhPDGF-BB + membrane. This study showed the reproducibility of the beneficial effects of rhPDGF-BB on bone regeneration, whereas the sites treated with a graft plus a membrane, but lacking rhPDGF-BB, resulted in little to no vertical bone regeneration. Although the results demonstrated that the two regenerative procedures that included the addition of rhPDGF-BB resulted in a significant bone regeneration and large bone-to-implant contact, no additional benefit could be observed for the combined treatment of eHAC block + rhPDGF-BB + membrane.

The question of whether a membrane is needed, beneficial, or inhibitory to rhPDGF-mediated regeneration was further evaluated by Al-Hazmi and co-workers [37]. The aim of this tomographic study was to assess the efficacy of using rhPDGF and xenograft with or without CM for GBR around immediate implants with dehiscence defects in dogs. After 16 weeks, the grafted sites were assessed for buccal bone thickness (BBT), buccal bone volume (BBV), vertical bone height (VBH), and bone-to-implant contact (BIC). All assessments were the greatest for the rhPDGF plus xenograft without a membrane. In other words, the membrane reduced the amount of rhPDGF mediated bone regeneration around the implants. The authors concluded that GBR around immediate implants with dehiscence defects using rhPDGF and xenograft alone resulted in higher BBT, BBV, VBH, and BIC than when performed in combination with CM.

### 2.5. Histological Outcomes in Alveolar Ridge Preservation

Studies histologically evaluating the use of rhPDGF-BB for the augmentation of extraction sockets for implant placement started with a case series by Nevins et al. [38]. In this study, eight cases with buccal wall extraction defects were treated with a mineralized collagen bone substitute (MCBS) combined with PDGF-BB. After 4 or 6 months of healing, core specimens for the grafted sites were harvested at site re-entry for implant placement. Histologic analysis revealed new bone formation all over the extraction socket, with close contact between the new bone and MCBS remnants. Osteocytes and osteoblasts were also present, with robust osteoid lines. Histomorphometric analysis of the 4- and 6-months samples demonstrated comparable mean percentages of new bone (23.2% ± 3.2% vs. 18.2% ± 2.1%), residual MCBS (9.5% ± 9.1% vs. 17.1% ± 7.0%), and soft tissue (67.3% ± 11.6% vs. 64.7% ± 7.1%). Furthermore, McAllister compared rhPDGF-BB (0.3 mg/mL) combined with either ABB-C or b-TCP for the treatment of patients with 12 extraction sockets [22]. Histomorphometric analysis of core biopsies after 3 months of healing showed that both b-TCP and ABB-C were similar regarding mean vital bone formation (21% vs. 24%) and residual graft particles (24% and 17%). These studies were followed up with a case series by Wallace et al. who performed a ridge preservation procedure using either mineralized allograft alone or in combination with rhPDGF-BB [23]. Histophorphometric studies after 4 months showed that the rhPDGF-BB group resulted in less residual allograft and increased vital bone percentage. In addition, Geurs et al. treated extraction sockets in 41 patients either with collagen plug (control), FDBA + b-TCP + collagen plug, FDBA + b-TCP + platelet rich plasma (PRP) + collagen plug, or FDBA + b-TCP + rhPDGF-BB + collagen plug for 8 weeks [24]. Histomorphometric analysis showed that the group in which rhPDGF was used showed new woven bone formation, less organic matrix, and the least amount of residual bone graft. Recently, a study by Mendoza-Azpur et al. tested the use of rhPDGF-BB for ridge preservation following extraction [25]. Patients were grafted with either anorganic bovine bone and collagen + rhPDGF-BB, anorganic bovine bone and collagen alone, or did not receive any biomaterial (control) after tooth extraction. After 4 months of healing, histologic and histomorphometric analysis showed no statistically significant differences between the groups regarding osteocytes and osteoblasts count and new mineralized tissue formation, while the rhPDGF-BB group showed a significantly higher number of blood vessels as well as Musashi-1 positive cells.

### 2.6. Histological Outcomes in Guided Bone Regeneration

The effect of PDGF on guided bone regeneration procedures has been tested in numerous studies on humans and animals. Simion et al. treated vertical alveolar ridge defects in two patients; one of them received deproteinized bovine bone block infused with rhPDGF-BB, and the other one was treated with deproteinized bovine bone graft embedded in a rhPDGF-BB-socked collagen matrix [26]. Five months after implant placement, core biopsies were harvested for histological analysis. Both grafting modalities resulted in new bone formation. The histologic influence of PDGF on lateral alveolar ridge augmentation has been reported in a study by De Angelis et al., where a patient was treated with equine block in combination with rhPDGF-BB [27]. After 3 months, histologic analysis of the core biopsy demonstrated replacement of the bone graft with a new organized bone that was dense with marrow and infiltrating blood vessels.

Nevins et al. used blocks of equine eHAC alone (control) or in combination with rhPDGF-BB (test) for the treatment of bilateral, surgically-created, vertical bone defects in monkeys [28]. Four months after augmentation, histologic analysis showed that both treatments resulted in vertical ridge augmentation with newly formed bone found in close contact with residual matrix particles, the underlying basal bone, and the tenting screws. The PDGF group showed a tendency to increase augmentation relative to the control group; however, the results did not attain statistical significance. In another study, Nevins et al. treated eight patients with either bovine or equine matrix mixed with rhPDGF-BB (0.3 mg/mL) [29]. After a 5-month healing period, histologic studies revealed newly formed bone in close contact with the graft particles in both groups.

### 2.7. Histological Outcomes in Sinus Augmentation

Histologic evidence testing the regenerative potential of rhPDGF in sinus augmentation was performed in only one study by Nevins et al. [21]. The study reported using anorganic bovine bone mineral in combination with rhPDGF-BB in 13 lateral window maxillary sinus augmentations in 10 patients. After a 6- to 8-month healing period, histologic analysis of core specimens revealed the formation of well-vascularized lamellar bone and woven bone. Some samples demonstrated resorption of the graft particles, while others contained intact residual graft particles.

## 3. Conclusions

Histologic evaluation of biopsies remains one of the most sensitive techniques with which to assess tissue and cellular responses to treatment. Simply stated, histologic evaluation can evaluate responses to therapies, especially growth factors, and either positive or negative reactions that cannot be determined using radiographic or clinical assessments. Therefore, the authors sought to systematically review the histologic response to rhPDGF-BB to determine whether any adverse reactions to its use had been noted on a microscopic level, and whether it appeared to improve tissue regeneration as evaluated histologically in many different clinical indications. Several conclusions can be drawn from this review of histologic studies on rhPDGF-BB:No abnormal or unfavorable effects, such as increased inflammation, ankylosis, or tissue overgrowth, have been reported following the use of rhPDGF-BB, even at the tissue and cellular levels under careful microscopic evaluation following nearly 30 years of histologic evaluation in animal models and many different human clinical indications.The osteopromotive effects of rhPDGF are present histologically as early as two weeks following treatment in animal models.The use of rhPDGF has been most thoroughly evaluated in humans histologically in periodontal infrabony and furcation defects. These studies have consistently found that true periodontal regeneration of cementum, PDL, and bone can be achieved using rhPDGF with allograft, xenograft, and, to a lesser extent, beta-TCP. In contrast, open-flap surgical debridement results in a long junctional epithelium.The histologic effects of rhPDGF have also been evaluated in numerous other clinical indications including peri-implant defects, GBR, extraction sockets, and, to a lesser extent, sinus grafts. In all cases, there was either a statistical improvement in bone regeneration (mostly) or a strong trend towards more bone in rhPDGF-treated sites. Such consistency of observations, despite the differences that might be expected across investigators, indications, models, and surgical techniques, etc., is striking.The use of collagen membranes is contra-indicated when using rhPDGF in intrabony defects and other sites where graft containment and stabilization can be achieved without the use of membranes. If a membrane is indicated for graft containment or stabilization, one should consider a membrane that is not cell occlusive (such as perforated PTFE) when using rhPDGF.

## Figures and Tables

**Figure 1 medicina-59-00676-f001:**
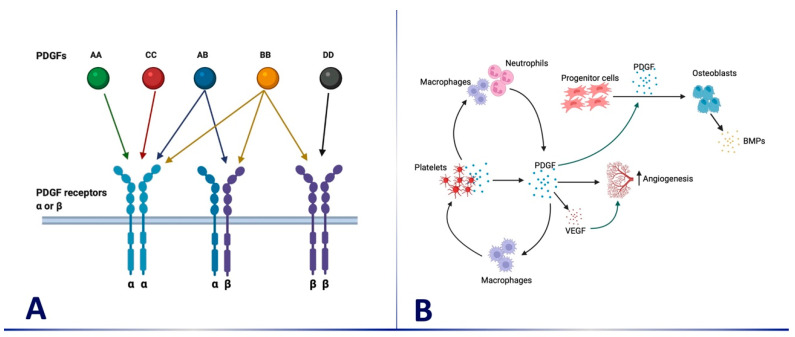
(**A**) PDGF 4 isoform homodimers and heterodimers binding on PDGF receptors for dimers on a cell surface (α and/or β). PDGF-AA, -BB, and -CC bind to PDGFR α/α, PDGF-AB, -BB, -CC, and -DD bind to PDGFR α/β, while PDGF-BB and DD bind to PDGFR β/β; (**B**) At injury sites, platelets release α-granules containing PDGFs, which in turn induce chemotaxis and proliferation of a multitude of cells important in healing and bone regeneration. The activation of macrophages and neutrophils results in increased production of PDGF, which results in increased angiogenesis directly or as a result of increased vascular endothelial growth factor (VEGF). PDGF also induces differentiation of osteoblast progenitors to osteoblasts, resulting in increased production of bone morphogenetic proteins (BMPs) by osteoblasts.

**Figure 2 medicina-59-00676-f002:**
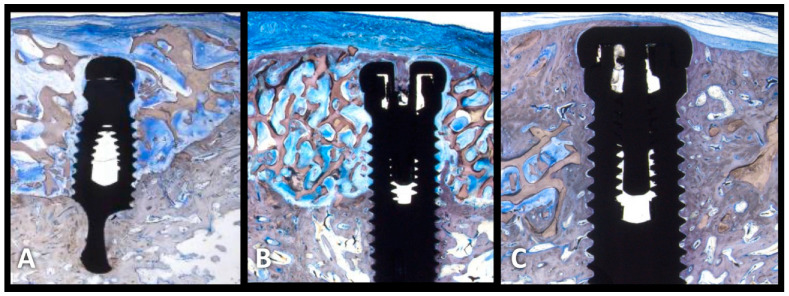
Vertical bone regeneration using rhPDGF + Bone block. (**A**) GBR using deproteinized bone block and collagen membrane. Bone block embedded in connective tissue and absence of new bone (4 months); (**B**) GBR using deproteinized bone block and collagen membrane and rhPDGF. New bone is present in the apical and coronal aspects of the implants, but the center is dominated by connective tissue (4 months); (**C**) GBR using deproteinized bone block with rhPDGF (no membrane). New bone is present throughout the entire length of the implants (4 months).

**Table 1 medicina-59-00676-t001:** Summary of the studies included in this review.

**Intrabony Defects**							
**Reference**	**Species/Model**	**Bone Graft Type/Carrier**	**Amt of rhPDGF: Saturated**	**Sample Size**	**Defect Type**	**Healing Type Histologically**	**Clinical Gains (Mean mm)**
Lynch et al., 1991 [13] PDGF in combination with IGF	Animal (Beagle dogs)	Methylcellulose gel	10 ng of I-PDGF-B in combination with IGF	13	Not reported	New Bone, new cementum, highly organized CT	Not reported
Wang et al., 1994 [14]	Animal (Mongrel dogs)	PDGF alone or in combination with ePTFE membrane	Not reported	6	Fenestration defects created into dentin	Increased fibroblasts, cemetoblasts, osteoblasts, perivacular cells, and endothelial cells proliferation	Not reported
Sculean et al., 1997 [15]	Animal (Monkey)	PDGF in gel	0.1 µg/mL	1	Surgically- created intrabony defects on either the mesial or distal aspect of teeth	New PDL, new cementum, new bone	Not reported
Nevins et al., 2003 [16]	Human	DFDBA	0.3 mg/mL, 1.0 mg/mL, and 5.0 mg/mL	9	Interproximal intrabony defects	New PDL, new cementum, new bone	PD reduction 6.42 ± 1.69 CAL gain 6.17 ± 1.94 Bone fill 2.14 ± 0.85
Ridgway et al., 2008 [17]	Human	β-TCP	0.3 mg/mL, and 1.0 mg/mL	8	Interproximal intrabony	New PDL, new cementum, new bone	PD reduction 4.6 ± 1.5 (0.3 mg/mL) 4.3 ± 1.5 (1 mg/mL) CAL gain 3.1 ± 1.8 (0.3 mg/mL) 3.2 ± 1.9 (1 mg/mL)
Shirakata et al., 2010 [18]	Animal (Beagle dogs)	β-TCP	0.3 mg/mL	4	Surgically created interproximal defects	New cementum, new bone	Bone formation 4.66 ± 0.7
**Furcation defects**							
**Reference**	**Species/Model**	**Bone Graft type/Carrier**	**Amt of rhPDGF: Saturated**	**Sample size**	**Defect type**	**Healing type Histologically**	**Clinical gains**
Nevins et al., 2003 [16]	Human	DFDBA	0.3 mg/mL, 1.0 mg/mL, and 5.0 mg/mL	9	Class III furcation defects	New PDL, new cementum, new bone	PD reduction 6.42 ± 1.69 CAL gain 6.17 ± 1.94 Bone fill 2.14 ± 0.85
Camelo et al., 2003 [19]	Human	DFDBA	0.5 mg/mL and 1.0 mg/mL	4	Class II furcation defects	New PDL, new cementum, new bone	PD reduction (Vertical 4.25) (Horizontal 3.5) CAL 3.75
Mellonig et al., 2009 [20]	Human	β-TCP	0.3 mg/mL	4	Class III furcation defects	New PDL, new cementum, new bone	PD reduction 4 CAL gain 2.86
**Alveolar ridge preservation**							
**Reference**	**Species/Model**	**Bone Graft type /Carrier**	**Amt of rhPDGF: Saturated**	**Sample size**	**Defect type**	**Healing type Histologically**	**Clinical gains**
Nevins et al., 2009 [21]	Human	MCBS	0.3 mg/mL	8	Buccal wall extraction defects	New bone formation	Not reported
McAllister et al., 2010 [22]	Human	Either ABB-C or β-TCP	0.3 mg/mL	11	Extraction sockets	Vital bone formation	Not reported
Wallace et al., 2013 [23]	Human	Mineralized allograft	0.3 mg/mL	30	Buccal wall extraction defects	Vital bone formation	Not reported
Geurs et al., 2014 [24]	Human	FDBA + β-TCP + Collagen plug	NR	41	Extraction sockets	New woven bone formation, less organic matrix	Not reported
Mendoza-Azpur et al., 2022 [25]	Human	Anorganic bovine bone and collagen	0.3 mg/mL	5	Extraction sockets	New mineralized tissue formation	Significant differences in the bucco-lingual width between the groups, favoring the PDGF group
**Guided bone regeneration**							
**Reference**	**Species/Model**	**Bone Graft type/Carrier**	**Amt of rhPDGF: Saturated**	**Sample size**	**Defect type**	**Healing type Histologically**	**Clinical gains**
Simion et al., 2007 [26]	Human	Deproteinized bovine bone block	NR	2	Vertical and horizontal ridge defects	New bone formation	Not reported
De Angelis et al., 2011 [27]	Human	Equine block	0.3 mg/mL	1	Horizontal ridge defect	New bone formation	Not reported
Nevins et al., 2012 [28]	Monkey	eHAC block	NR	6	Vertical bone defects	New bone formation	Not reported
Nevins et al., 2014 [29]	Human	Either with bovine or equine matrix	0.3 mg/mL	8	Large alveolar extraction defects	New bone formation	Not reported
**Sinus augmentation**							
**Reference**	**Species/Model**	**Bone Graft type/Carrier**	**Amt of rhPDGF: Saturated**	**Sample size**	**Defect type**	**Healing type Histologically**	**Clinical gains**
Nevins et al., 2009 [21]	Human	ABBM	0.3 mg/mL	10		Lamellar bone and woven bone formation	Not reported
**Soft tissue augmentation and recession coverage**							
**Reference**	**Species/Model**	**Bone Graft type/Carrier**	**Amt of rhPDGF: Saturated**	**Sample size**	**Defect type**	**Healing type Histologically**	**Clinical gains**
McGuire et al., 2009 [30]	Human	β-TCP	0.3 mg/mL	30	Miller Class II buccal gingival recession	New PDL New Cementum New Bone	Root coverage (90.8%) Recession depth reduction (−2.9 + 0.5 mm)
Simion et al., 2012 [31]	Human	Resorbable collagen matrix	Not reported	6	Insufficient soft tissue volume	Well organized epithelium and CT	Increased soft tissue volume
**Dental Implant-related Regeneration**							
**Reference**	**Species/Model**	**Bone Graft type/Carrier**	**Amt of rhPDGF: Saturated**	**Sample size**	**Defect type**	**Healing type Histologically**	**Clinical gains**
Simion et al., 2006 [32]	Canine	Deproteinized bovine bone block	Not reported	6	Vertical ridge defects	New bone formation	Not reported
Simion et al., 2009 [33]	Canine	eHAC block	Not reported	12	Vertical ridge defects	New bone formation	Not reported

β-TCP = Beta tricalcium phosphate; DFDBA = Demineralized freeze-dried bone allograft; MCBS = Mineralized collagen bone substitute; ABB-C = Anorganic bovine bone collagen; FDBA = Freeze-dried bone allograft; eHAC = equine hydroxyapatite collagen; ABBM = Anorganic bovine bone mineral; PDL = Periodontal ligament; CAL = Clinical attachment loss; PD = Pocket depth; CT = Connective tissue; PDGF = Platelet-derived growth factor

## Data Availability

Not applicable.

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
