# Peer review of "Histologic Evidence of Oral and Periodontal Regeneration Using Recombinant Human Platelet-Derived Growth Factor"

_medicina, 2023, doi:10.3390/medicina59040676_

Round 1
Reviewer 1 Report
The authors aim to summarize the histological findings following the use of Human Platelet - Derived Growth Factor in humans and animal models. The topic is interesting and this review helps to a better understanding of this topic for those interested in periodontal tissue regeneration, however, the manuscript is poorly organized and it need to be improved. To start with, the manuscript does not follow the style of the journal, and it lacks line numbers, so it is difficult to mention the specific points on each paragraph. There is a number of improvements that could be done, as for example, there no Table 1 (or at least I could not download it) , the abbreviations are not introduced properly, and specially, the figures are poor. Since the authors are also self cited, I think there must be unpublished data that could be shown to enrich the manuscript. The two figures showe are poorly described, the abbreviations inside the figure are not explained, they lack scales bars, and specially in Figure 2, I sugggest the authors should divide it into two, and add more data. On Figure 3, although looks good, it could be better explained not only in the figure legend but in the main text.
Author Response
Reviewer 1:
- The manuscript does not follow the style of the journal, and it lacks line numbers: The manuscript has been revised according to the journal’s style.
- There is a number of improvements that could be done, as for example, there no Table 1: A table was added to the manuscript
- The abbreviations are not introduced properly. This has been corrected and abbreviations were defined in the text and figures.
- The figures are poor. The figures were added to the manuscript text according the journal’s style. This might have compromised the resolution.
- Since the authors are also self cited, I think there must be unpublished data that could be shown to enrich the manuscript.
Thank you for your comment. We don’t have unpublished data that we are currently able to share here.
- The two figures showed are poorly described, the abbreviations inside the figure are not explained, they lack scales bars, and specially in Figure 2, I suggest the authors should divide it into two, and add more data. Figure legends were revised
- On Figure 3, although looks good, it could be better explained not only in the figure legend but in the main text.
- Thank you for your comment. The figure is now explained in more detail in the text.
Reviewer 2 Report
General Comments
This manuscript is meant to report on the Histologic evidence of oral and periodontal regeneration using the Recombinant Human Platelet–Derived Growth Factor. In its present form, considerable improvements in the write-up are necessary.
1. The Title:
i. The length is Ok.
ii. The title does present what's in the manuscript.
iii. The part “oral” can be omitted since oral tissues is a wider term that comprises teeth, buccal and palatal mucosa and other oral tissues apart from periodontal tissues
iv. The title should say it’s a review article otherwise, it seems like a case-control study
Recommendation: Please rephrase the title. Eg: Histological evidence of periodontal tissues regeneration using the Recombinant Human Platelet-Derived Growth Factor: A review
Abstract: i) Number of words is within the range of the journal requirement.
ii) It contains the aim of the research but it is not very clearly put.
iii) It does not give a rationale for carrying out this study.
iv) Please rephrase: a similar communication is needed focused on human histologic outcomes it is grammatically incorrect
Recommendation: The abstract can be improved.
Keywords: They are relevant except for biologics.
Introduction and the body:
i. The introduction describes briefly the area of interest, but largely it drifts the reader away from the purpose of the review. The first paragraph is all about the definition of terms. How do
ii. The motivation for the review has not been clearly put forward
iii. The rationale for review is not clearly elaborated.
iv. Minimum grammatical and spelling errors noted.
v. Citation of references used not made for some sentences or phrases
vi. The rest of body does review various relevant studies.
vii. The authors use abbreviation of some words without giving long-form e.g LJE, OFD etc. Please provide a long form before using the abbreviation.
Recommendation: Please improve the introduction part
Conclusion:
i. The conclusion is very lengthy.
ii. The conclusion does not answer the objectives. Some of the points eg paragraph 2 is out of context in relation to the aim and what the authors have reviewed.
iii. Authors have failed to summarize main points of the review.
Recommendation: Please improve the conclusion.
Structure and length: i) It is a moderately lengthy paper
ii) The article is not well-organized nor well-balanced as far as introduction and conclusion are considered.
iii) It has some relevant information.
Recommendation: the article requires to be re- organized.
Logic:
i. The article has been written clearly especially the body. Rest of the introduction and conclusion misses the logic
ii. There is some violation of grammar There is logical consistency in most of the paragraphs throughout the text.
Recommendation: Please improve the introduction and conclusion
Figures: i) Figure 2 is poorly describe
English: i) The English used in the article is good enough to convey the scientific meaning correctly. Though there are minimum grammatical errors
Recommendation: there is room for improving the English used.
References: i) Most of the references are more than a decade old
Recommendation: Please use references which are latest, so the review can have more relevance based on current advancement in the field of medicine.
Author Response
- The part “oral” can be omitted since oral tissues is a wider term that comprises teeth, buccal and palatal mucosa and other oral tissues apart from periodontal tissues. The authors choose to add Oral to the title because the article summarizes the histologic evidence of regeneration using PDGF in periodontal as well as ridge augmentation, site preservation, peri-implant sites and sinus augmentation.
- The title should say it’s a review article otherwise, it seems like a case-control study. A review word was added to the title.
Abstract:
- It does not give a rationale for carrying out this study.
Thank you for your comment. We understand your concern. This is not a study. It is a review of the histologic evidence of PDGF.
- Please rephrase: a similar communication is needed focused on human histologic outcomes. This sentence was rephrased.
Keywords:
- They are relevant except for biologics. The word biologics was deleted
Introduction and the body:
- The introduction describes briefly the area of interest, but largely it drifts the reader away from the purpose of the review. The first paragraph is all about the definition of terms. The authors wanted to provide the reader with information about PDGF and its biologic function and influence on tissue regeneration.
- The motivation for the review has not been clearly put forward.
Thank you. The aim of the review is more clearly stated.
- Minimum grammatical and spelling errors noted. Thank you. Several corrections were made and others can be corrected at the proofreading
- Citation of references used not made for some sentences or phrases
Thanks for brining that to our attention. All citations were reviewed for correctness.
- The authors use abbreviation of some words without giving long-form e.g LJE, OFD etc. Please provide a long form before using the abbreviation. This was corrected. All abbreviations were defined.
Conclusion:
- The conclusion is very lengthy. The manuscript aimed to summarize all reports that investigated the histologic evidence of regenerative potentials of PDGF. Hence, the conclusion had to be sufficient.
- The conclusion does not answer the objectives. Some of the points e.g. paragraph 2 is out of context in relation to the aim and what the authors have reviewed.
Thank you. We believe paragraph 2 is essential in providing some background to the PDGF.
Structure and length:
- The article is not well-organized nor well-balanced as far as introduction and conclusion are considered.
Thank you. Several corrections were made to the article to make the structure of the article more balanced.
Figures:
- Figure 2 is poorly described.
Thank you. The figure was histologic slide from previous work of one of the authors. The figure was removed to avoid confusion.
References:
- Most of the references are more than a decade old. The aim of this review was to summarize all studies that reported on the histologic evidence of the regenerative potential of PDGF since the molecule was used in periodontal and oral regenerative procedures.